# Systemic Dendrimer-Peptide Therapies for Wet Age-Related Macular Degeneration

**DOI:** 10.3390/pharmaceutics15102428

**Published:** 2023-10-05

**Authors:** Tony Wu, Chang Liu, Rangaramanujam M. Kannan

**Affiliations:** Center for Nanomedicine, Wilmer Eye Institute, Johns Hopkins University School of Medicine, Baltimore, MD 21287, USA; twu29@jhu.edu (T.W.); cliu188@jhu.edu (C.L.)

**Keywords:** integrin, peptide conjugates, AMD, hydroxyl PAMAM dendrimer, targeted delivery, neovascularization

## Abstract

Wet age-related macular degeneration (AMD) is an end-stage event in a complex pathogenesis of macular degeneration, involving the abnormal growth of blood vessels at the retinal pigment epithelium driven by vascular endothelial growth factor (VEGF). Current therapies seek to interrupt VEGF signaling to halt the progress of neovascularization, but a significant patient population is not responsive. New treatment modalities such as integrin-binding peptides (risuteganib/Luminate/ALG-1001) are being explored to address this clinical need but these treatments necessitate the use of intravitreal injections (IVT), which carries risks of complications and restricts its availability in less-developed countries. Successful systemic delivery of peptide-based therapeutics must overcome obstacles such as degradation by proteinases in circulation and off-target binding. In this work, we present a novel dendrimer-integrin-binding peptide (D-ALG) synthesized with a noncleavable, “clickable” linker. In vitro, D-ALG protected the peptide payload from enzymatic degradation for up to 1.5 h (~90% of the compound remained intact) in a high concentration of proteinase (2 mg/mL) whereas ~90% of free ALG-1001 was degraded in the same period. Further, dendrimer conjugation preserved the antiangiogenic activity of ALG-1001 in vitro with significant reductions in endothelial vessel network formation compared to untreated controls. In vivo, direct intravitreal injections of ALG-1001 and D-ALG produced reductions in the CNV lesion area but in systemically dosed animals, only D-ALG produced significant reductions of CNV lesion area at 14 days. Imaging data suggested that the difference in efficacy may be due to more D-ALG remaining in the target area than ALG-1001 after administration. The results presented here offer a clinically relevant route for peptide therapeutics by addressing the major obstacles that these therapies face in delivery.

## 1. Introduction

Millions of elderly patients face the risk of vision loss due to age-related macular degeneration (AMD) and approximately 10% of these patients will develop wet AMD [1,2,3,4]. Wet AMD is a late-stage event in the complex pathological evolution of AMD, in which choroidal neovascularization pushes blood vessels from the choroid through the Bruch’s membrane to displace or destroy the retinal pigment epithelium (RPE) [5,6,7]. A prominent factor in the pathogenesis of wet AMD is the elevated expression of vascular endothelial growth factor (VEGF) in the eye, which encourages the growth of new blood vessels [8,9,10,11]. As a result, most current standards of care for wet AMD target extracellular VEGF directly through intravitreal injections of anti-VEGF antibodies such as aflibercept [12,13,14,15,16]. However, a significant portion of patients (~1/3) do not respond to these therapies [17,18,19] and vision can still decline despite optimal patient adherence to the treatment regimen [20].

Other therapeutic modalities such as integrin-binding peptides are being developed to halt the progression of wet AMD in patients [21,22,23]. These peptide antagonists bind strongly to cell surface integrins such as αVβ3, α5β1, and α5β3 and inhibit the downstream signaling of these integrins [24,25,26]. Specifically, these integrin antagonists reduce the activation of the ERK and PI3K/Akt pathways, which in turn attenuate the expression of a variety of proinflammatory and proangiogenic cytokines such as VEGF-A, TNF-α, and IL1β [27,28,29,30]. Luminate (or ALG-1001) is one such integrin-binding peptide that has proven successful in halting neovascularization and is currently undergoing clinical trials for applications in AMD and in diabetic macular edema (DME) [28,31,32]. However, peptide-based antagonists suffer from a wide range of delivery challenges including rapid enzymatic degradation and renal clearance, and lack of cell-targeting [33,34]. Thus, these therapies are currently restricted to intravitreal injections, which not only limits their availability to patients in less developed countries [35] but also carry risks of endophthalmitis, elevated intraocular pressures (IOP), and irritation [36,37,38,39].

Systemic treatments for AMD may enable treatments during the early stages of the disease [40] while bypassing the challenges associated with intravitreal injections. In recent years, hydroxyl-terminated, neutrally charged dendrimers have emerged as a promising biocompatible platform for drug delivery, with multiple products in clinical trials. In numerous animal model and human studies, dendrimer conjugation of small molecule drugs improved their safety profile and increased their efficacy in the target area [40,41,42,43,44,45]. The inherent ability of the dendrimer to be taken up by reactive macrophages and microglia allows for a higher local concentration to be achieved while the rapid clearance of dendrimer conjugates from the blood reduces unnecessary exposure in nontargeted cells and tissues [46,47,48]. Recent phase 2a clinical trials of hydroxyl PAMAM dendrimer-NAC conjugates in hospitalized severe COVID-19 patients showed that a single intravenous dose of this conjugates led to an 82% survival, the attenuation of hyperinflammation, and a significant reduction in blood biomarkers of brain injury associated with COVID, compared to the standard of care. This suggests that the safety and inflammatory cell-targeting viability of this approach can be leveraged clinically to improve patient outcome [49]. Further, dendrimer conjugation of peptides have also been employed to enable superior cell surface binding because multivalency prevents the binding of the SARS-CoV-2 spike protein with the ACE2 receptor at 1–3 magnitudes lower concentration compared to the free peptide [50]. In this work, we demonstrate that the advantages conferred to small molecule drugs by dendrimer conjugation can be extended to small-molecule biologics [51,52,53]. Dendrimer conjugation to ALG-1001 peptide preserves the activity of the peptide, increases its stability, prolongs its residence time in the target tissue, and offers systemic administration as an alternative route to intravitreal injections, thereby expanding the availability of the therapy worldwide.

## 2. Materials and Methods

### 2.1. Chemicals and Reagents

Hydroxyl-terminated, ethylenediamine-core PAMAM dendrimer (generation 6, pharmaceutical grade) in methanol solution was purchased from Dendritech (Midland, MI, USA). Prior to use, the dendrimer solution was evaporated on a rotary evaporator. The dialysis membrane (MWCO 1 kDa) was purchased from Spectrum Chemicals (New Brunswick, NJ, USA). ALG-1001 and ALG-1001 modified with azide linker were purchased from Bio-Synthesis (Lewisville, TX, USA). Deuterated solvents (DMSO-*d*_6_), methanol (CD_3_OD), and water (D_2_O) were purchased from Sigma-Aldrich. Proteinase K stock solution and Dulbecco’s modified Eagle medium (DMEM, low glucose with L-glutamine) were purchased from ThermoFisher (Waltham, MA, USA). Human umbilical vein endothelial cells and the required media kit were purchased from Lonza (Basel, Switzerland). Phenol Red-free Matrigel was purchased from Corning (Tewksberry, MA, USA).

### 2.2. Instrumentation

#### 2.2.1. Nuclear Magnetic Resonance

A Bruker 500 MHz spectrometer was used to obtain ^1^H NMR spectra at ambient temperatures. Chemical shifts are reported in parts per million relative to tetramethylsilane, which was used as an internal standard, and the residual protic solvent peaks were used for chemical shifts’ calibration. DMSO-*d*_6_ (δ = 2.50 ppm). The resonance multiplicity in the spectra is indicated as “s” (singlet), “d” (doublet), “t” (triplet), and “m” (multiplet). Broad resonances are expressed by “b”.

#### 2.2.2. High-Performance Liquid Chromatography

Waters HPLC (Milford, MA, USA) equipped with a 1525 binary pump, and in-line degasser AF, a 717 plus autosampler, and a 2998 photodiode array detector interfaced with Waters Empower 2 software was used to determine the purities of compounds. The column was a Waters Symmetry C18 reversed-phase column with a particle size of 5 µm, a length of 25 cm, and an internal diameter of 4.6. mm. The chromatograms were monitored at 210, 650, and 530 nm using the photodiode array (PDA) detector. The analysis was performed with a gradient flow starting at 95:5 (H_2_O/ACN) increasing to 50:50 (H_2_O/ACN) in 30 min and returning to 95:5 (H_2_O/ACN) in 10 min at a flow rate of 1 mL/min.

### 2.3. Synthesis of Dendrimer Conjugates

#### 2.3.1. Synthesis of Dendrimer-Hexyne

5-Hexynoic acid and DMAP were added to a solution of D-OH in anhydrous DMF and stirred at RT for 15 min. EDC·HCl was added to the resulting clear solutions in 3 equal portions and the solution was stirred overnight at room temperature. The reaction mixture was purified by dialysis through a 2 kDa MW cut-off cellulose dialysis membrane against DMF with solvent change at 8 h intervals. After 24 h, the mixture was dialyzed against water for 24 h with solvent change at 12 h intervals. The final aqueous solution was lyophilized to get the product as a white solid.

^1^H NMR (500 MHz, DMSO) δ 8.06–7.79 (m, internal amide 510H), 4.73 (s, surface OH, 232H), 4.01 (s, ester linked, 22H), 3.40 (d, dendrimer-CH_2_), 3.33 (d, dendrimer-CH_2_), 3.18–3.11 (m, dendrimer-CH_2_), 2.64 (s, dendrimer-CH_2_), 2.43 (s, dendrimer-CH_2_), 2.20 (s, dendrimer-CH_2_), 1.68 (t, acetylene, 30H). Retention time: 19.58 min

#### 2.3.2. Synthesis of BOC-GABA-D-Hexyne

D-hexyne was dissolved in anhydrous DMF, and BOC-GABA-OH and DMAP were added. The solution was stirred for 15 min before the addition of EDC·HCl in 3 equal, separate portions. The solution was stirred overnight, purified by dialysis, and lyophilized to get the product as a white solid.

^1^H NMR (500 MHz, DMSO-d_6_) δ 8.06–7.79 (m, internal amide 510H), 4.73 (s, surface OH, 199H), 4.06 (s, ester linked, 40H), 3.40 (d, dendrimer-CH_2_), 3.33 (d, dendrimer-CH_2_), 3.18–3.11 (m, dendrimer-CH_2_), 2.64 (s, dendrimer-CH_2_), 2.43 (s, dendrimer-CH_2_), 2.20 (s, dendrimer-CH_2_), 1.68 (t, acetylene, 27H), 1.36 (s, BOC, 68H).

#### 2.3.3. Synthesis of GABA-D-Hexyne

The deprotection of BOC-GABA-D-hexyne was performed under anhydrous conditions. The compound was placed in a round bottom flask, and anhydrous DCM was added under a nitrogen atmosphere. The solution was constantly stirred and sonicated to form a cloudy, sticky suspension. TFA was then added to the suspension in a 4:1 ratio (DCM:TFA), and the solution was stirred overnight. DCM was then evaporated using a rotary evaporator. TFA was removed by repeatedly diluting the reaction mixture with methanol and evaporating the resultant solution. The product was then placed under high vacuum for 3 h and used without further purification.

#### 2.3.4. Synthesis of Cy5-D-Hexyne

GABA-D-hexyne was dissolved in anhydrous DMF, followed by DIPEA, and finally the addition of Cy5 NHS ester. The reaction was stirred overnight, and the reaction mixture was dialyzed through a 2 kDa membrane against DMF for 24 h. The mixture was then dialyzed against water for an additional 24 h and lyophilized to obtain a solid, blue product.

^1^H NMR (500 MHz, DMSO-d_6_) δ 8.06–7.79 (m, internal amide 510H), 7.35 (m, Cy5 H), 7.25 (m, Cy5 H), 7.05 (m, Cy5 H), 6.6 (m, Cy5 H), 6.3 (m, Cy5 H), 4.73 (s, surface OH, 199H), 4.06 (s, ester linked, 40H), 3.40 (d, dendrimer-CH_2_), 3.33 (d, dendrimer-CH_2_), 3.18–3.11 (m, dendrimer-CH_2_), 2.64 (s, dendrimer-CH_2_), 2.43 (s, dendrimer-CH_2_), 2.20 (s, dendrimer-CH_2_), 1.68 (t, acetylene, 27H). Retention time: 18.01 min

#### 2.3.5. Synthesis of D-ALG1001 and Cy5-D-ALG1001

ALG-1001 was dissolved in ultrapure water and added to an aqueous solution of D-hexyne for unlabeled conjugates. ALG-1001-Cy3 was dissolved in ultrapure water and added to an aqueous solution of Cy5-D-hexyne for dual-labeled conjugates. We then added a solution of copper sulfate and stirred the solution for 10 min at RT before adding sodium ascorbate. For purification, both reactions were left at room temperature overnight then dialyzed against water for 24 h. Each solution was lyophilized to obtain a powder product.

^1^H NMR (500 MHz, DMSO-d_6_) of D-ALG: 8.12–7.78 (m, internal amide H), 4.45–4.06 (m, peptide α carbon), 4.00 (s, ester linked, 37H), 3.78 (m, polyethylene glycol H), 3.40 (d, dendrimer-CH_2_), 3.33 (d, dendrimer-CH_2_), 3.18–3.11 (m, dendrimer-CH_2_), 2.64 (s, dendrimer-CH_2_), 1.64–1.59 (m, GABA linker-CH_2_ and peptide side chain). Retention time: 16.60 min

^1^H NMR (500 MHz, DMSO-d_6_) of Cy5-D-ALG-Cy3: 8.12–7.78 (m, internal amide H), 7.01 (m, aromatic 5H), 6.59 (s, GABA amide H, 10H), 4.73 (s, surface OH, 233H), 4.06 (s, ester linked, 16H), 3.40 (d, dendrimer-CH_2_), 3.33 (d, dendrimer-CH_2_), 3.18–3.11 (m, dendrimer-CH_2_), 2.64 (s, dendrimer-CH_2_), 1.64–1.59 (m, GABA linker-CH2, 6H), 1.36 (s, Boc group, 20H). Retention time: 17.99 min

### 2.4. In Vitro Stability under Enzymatic Degradation

Proteinase K stock solution was purchased from ThermoFisher and used as is. ALG-1001 and D-ALG solutions were prepared at a concentration of 2 mg/mL, and proteinase K was added to each solution such that the final concentration of proteinase K was 2 mg/mL. The mixture was incubated at 37 °C and at set time points, 100 µL of the mixture was removed and analyzed through HPLC. To determine compound degradation, the integrations of peaks at elution times associated with ALG-1001 and D-ALG were used and normalized to the injected analyte peak at the 0 h time point.

### 2.5. Cell Culture

HUVEC cells were obtained from Lonza and cultured in EGM-2 endothelial cell growth medium (Lonza). Murine macrophages (RAW264.7) between passages 5–9 were cultured in Dulbecco’s modified Eagle’s medium (DMEM, Life technologies, Grand Island, NY, USA) supplemented with 10% fetal bovine serum (Invitrogen Corp., Carlsbad, CA, USA) and 1% penicillin/streptomycin (Invitrogen Corp., Carlsbad, CA, USA). All cells were maintained at 37 °C and 5% CO_2_ with a humidified incubator.

### 2.6. In Vitro Evaluation of D-ALG Efficacy

#### 2.6.1. Vessel Formation Assay

First, 96-well plates were coated with 75 µL of Matrigel and left at room temperature for 15 min before being placed into 37 °C incubators for an additional 30 min. Then, D-ALG1001 was dissolved at twice the desired concentration and 50 µL of the drug solution was added to the wells. HUVEC cells were then added to each well at a density of 70,000 cells/cm^2^. Live-cell images were taken at 12 h for analysis.

#### 2.6.2. Wound-Healing Assay

HUVEC cells were seeded at a density of 5 × 10^4^ cells per well in 24-well plates and set aside for at least 72 h for a uniform monolayer of cells to form. Cells were treated with D-ALG1001 and ALG-1001 for 24 h. A centimeter-long scratch was introduced to the cell monolayer with the tip of a 200 µL pipette tip. Images were taken on a Nikon

#### 2.6.3. VEGF Activation for Western Blotting and PCR

HUVEC cells were seeded at a density of 5 × 10^4^ cells per well in 24-well plates 24 h before treatment. Cells were treated with D-ALG1001 and ALG-1001 for 24 h before activation with VEGF for 5 min. Cells were collected and lysed for Western blotting using T-Per buffer (ThermoFisher) supplemented with PhosStop and proteinase inhibitor cocktail. Cells were lysed with Trizol and processed as previously described for qPCR analysis.

#### 2.6.4. In Vitro Inflammation Model

RAW264.7 cells were seeded at a density of 1 × 10^5^ cells per well in 12-well plates 48 h before treatment. Cells were incubated with D-ALG1001 and ALG-1001 for 24 h, the treatment media was aspirated, then LPS at a concentration of 10,000 endotoxin units/mL was added to stimulate inflammatory response. Samples were collected 3 h after LPS stimulation for qPCR analysis.

### 2.7. In Vivo Evaluation of D-ALG at Attenuating Choroidal Neovascularization

#### 2.7.1. In Vivo Laser CNV Rat Models

All animal procedures were approved by the Johns Hopkins Animal Care and Use Committee. Brown Norway rats between 8 and 12 weeks old were obtained and housed at constant temperature and humidity (20 ± 1 °C, 50 ± 5% humidity). Animals were anesthetized through intraperitoneal injections of a ketamine/xylazine cocktail (ketamine 50 mg/kg and xylazine 10 mg/kg). Pupils were dilated with topical 2.5% phenylephrine hydrochloride solution followed by 0.5% tetracaine hydrochloride solution. To induce CNV formation, four equally spaced lesions were created in the Bruch’s membrane with a built-in laser system on the Micron III SLO. Laser power was set to 240–250 W with a duration of 70 ms. A gonio ophthalmic solution was applied postoperatively to prevent the eyes from drying and forming cataracts.

On the day of CNV induction (Day 0), D-ALG1001 and ALG1001 was administered intraperitoneally at a dose of 150 µg on a peptide basis. Subsequent doses were administered every 4 days. At days 7 and 14, mice were sacrificed and enucleated. Eyes used for CNV image were fixed in 10% formalin for 1 h. Eyes used for qPCR, ELISA, and Western blotting were immediately stored at −80 °C until use.

#### 2.7.2. Tissue Preparation for Western Blotting and qPCR

In brief, 500 µL of T-Per supplemented with PhosStop and proteinase inhibitor cocktail was added to tubes containing dissected choroid. A scoop of 1.6 mm steel homogenization beads was added to each sample and then placed in TissueLyserLT (Qiagen, Germantown, MD, USA) at an oscillation frequency of 50 cycles/s for 15 min.

For qPCR, 200 µL of Trizol was added to tubes containing choroid and a scoop of 1.6 mm steel homogenization beads was added. The samples were placed in TissueLyserLT at an oscillation frequency of 50 cycles/s for 15 min.

#### 2.7.3. Pathway Activation Analysis with Western Blot and ELISA

In brief, the concentration of protein was determined with the BCA protein assay kit (Thermo Scientific, Rockford, IL, USA). Equal amounts of protein were denatured and resolved on 4–15% TGX gels (Bio-Rad, Hercules, CA, USA). Gels were transferred to nitrocellulose membranes, blocked with 3% BSA at RT for 1 h and probed for GAPDH, FAK, pFAK, MAPK, and pMAPK at 4 °C overnight. Membranes were washed thrice before incubation with HRP-conjugated secondary antibodies followed by incubation with chemiluminescent substrate for visualization.

For protein extracted from tissues, total FAK and FAK (Phospho) [pY397] ELISA kits (Invitrogen) were used to quantify the level of protein expression, and the measured protein expression was normalized to the total protein amount of each sample as determined from BCA assays.

#### 2.7.4. qPCR Analysis

In brief, 100 µL of chloroform was added to the Trizol suspension, and the aqueous phase was separated with a centrifuge at 10K RPM at 4 °C for 15 min. Then, 400 µL of 2-propanol was added to the aqueous solution and spun again to pellet the RNA. The RNA was washed with 70% ethanol solution, pelleted again, and resuspended in DEPC water.

To convert RNA to complementary DNA, 2 µg of RNA was converted using a High-Capacity cDNA Reverse Transcription Kit (Applied Biosystems, Foster City, CA, USA). Samples were analyzed using StepOne Plus real-time PCR system (Applied Biosystems) with SYBR Green reagent (ThermoFisher Scientific). Relative expression was quantified with ΔΔCt calculations normalized to controls. Primers for GAPDH was obtained from Bio-Rad Laboratories (Hercules, CA, USA). Primers for TNFα (forward: CCA GTG TGG GAA GCT GTC TT; reverse: AAG CAA AAG AGG AGG CAA CA), IL1β (forward: AGC TTC AAA TCT CGA AGC AG; reverse: TGT CCT CAT CCT GGA AGG TC) were purchased from Integrated DNA Technologies (Coralville, IA, USA).

#### 2.7.5. Biodistribution

For biodistribution studies, a dose of Cy5-D-ALG1001-Cy3 and ALG-1001-Cy3 at 150 µg/100 µL was administered to each animal intraperitoneally on the day of CNV induction (day 0). Animals were sacrificed at 1-, 2-, 3-, and 4-day time points and enucleated.

#### 2.7.6. Imaging

After fixation, the posterior segment of the eye was dissected out and the retina separated from the choroid. Tissues stained with FITC-labeled isolectin (GS IB_4_) (Life Technologies, Eugene, OR, USA) for blood vessels and monocytes. Eyes were mounted by introducing four radial relaxation cuts. Samples for biodistribution were imaged under a confocal 710 microscope (Carl Zeiss, Oberkochen, Germany). Samples for CNV quantification were imaged with an Axiovert phase contrast microscope. All images were processed with ImageJ.

#### 2.7.7. Statistical Analysis

Data were presented as means ± SEM and analyses were performed in GraphPad Prism (version 9; La Jolla, CA, USA). Treatment groups across time point or doses were analyzed through an analysis of variance (ANOVA) tests. Significant differences among single groups were determined with Student’s *t* tests: * *p* < 0.05, ** *p* < 0.01 and *** *p* < 0.001.

## 3. Results

### 3.1. Synthesis and Characterization of D-ALG1001 Intermediates and Conjugates

We utilized a high-yield click reaction to efficiently attach the ALG-1001 peptide to the dendrimer platform under mild conditions, which is amenable to preserving the structural stability and activity of the peptide (Figure 1A) [54,55,56,57,58,59,60]. First, we modified the dendrimer surface functional groups with a hexynoic acid linker and confirmed the modification by ^1^HNMR with the presence of 20 protons at 4.0 ppm and 1.7 ppm (Figure 1B). The dendrimer surface was minimally modified to preserve its near-neutral charge and its inherent ability to penetrate tissues.

For biodistribution studies using Cy5 fluorophore, the dendrimer surface was further modified with GABA-Boc linkers. The presence of additional protons at 4.0 ppm, 1.7 ppm, and 1.2 ppm confirmed the modification. The resulting intermediate was deprotected and the free amine was used to couple with Cy5 ester to obtain a fluorescently labeled dendrimer (Appendix A).

ALG-1001 peptide was purchased with a short polyethylene glycol (PEG) azide linker attached to the C terminus and used without further preparation. For biodistribution studies, the N terminus of the peptide was also modified with Cy3 fluorophore for tracking. A copper (I)-catalyzed alkyne-azide click (CuAAC) reaction was used to attach ALG-1001 to dendrimers, and ^1^HNMR spectra confirmed the attachment of ALG-1001 by showing the presence of the peptide protons.

### 3.2. In Vitro Stability under Enzymatic Degradation

Coincubation of ALG-1001 and proteinase K, a broad acting protease, resulted in a rapid degradation of free ALG-1001 with about 50% degraded at 30 min and 90% at 90 min. On the other hand, the dendrimer conjugation of the ALG-1001 peptide coincubated with proteinase K did not result in HPLC peak shifts in retention time or produce additional peaks indicating the degradation of the compound at incubation times shorter than 90 min. The conjugation conferred resistance to enzymatic degradation possibly due to steric hinderance from the dendrimer carrier. Only around 10% of D-ALG was degraded at 90 min (Figure 2). This resistance toward enzymatic degradation, even at a high enzyme concentration (2 mg/mL) suggests that dendrimer conjugation increases the stability and may increase the circulation time of intact peptides in vivo by helping it escape degradation pathways. The concentration used (2 mg/mL) was about 20–40× higher than the concentration used for routine protease digestion protocols (50–100 µg/mL). The 2 mg/mL concentration was also 5× higher concentration than the 400 µg/mL used in assessing the stability of NF55/pDNA nanoparticles in which the compound resisted degradation for 3 h at most [61].

The generation-6 PAMAM dendrimers conjugates had a blood circulation time of ~4–8 h, reaching target cells over this time duration [46,62]. Therefore, the dendrimer conjugation may enable peptides to be largely delivered intact into target cells, at levels above what could be expected for free peptides, enabling us to address a major roadblock in the clinical translation of peptide-based therapeutics [63,64,65].

### 3.3. In Vitro Vessel Formation Assay

To compare the dose–response efficacy of free and D-ALG in a physiologically relevant model, we utilized an in vitro model of blood vessel formation [66]. HUVEC cells were treated with a gradient concentration of D-ALG and ALG-1001 for 24 h at three different magnitudes of doses. The cells were then seeded on Matrigel after which cells naturally migrate to form blood vessel-like tubule structures. The images were analyzed with the Angiogenesis Analyzer plug-in on ImageJ to extract relevant metrics that measure the connectivity and integrity of tube networks (Figure 3A and Appendix A).

While lower doses were not effective, cells treated with 1 mM of D-ALG and ALG-1001 exhibited an increased disruption in tube formation with more isolated segments and a smaller area or meshes enclosed by tubes. Interestingly, the trend suggested dendrimer conjugation may increase the efficacy of ALG-1001 at disrupting network formation. HUVECs treated with 1 mM of D-ALG had fewer points of intersection between vessels (junctions), fewer connected segments, and more isolated segments compared to those treated with 1 mM of ALG-1001.

### 3.4. Wound-Healing Assay

In addition to cell morphological and motility changes measured by tube formation assays, we investigated the activity of D-ALG and ALG-1001 in a wound-healing assay. Previous studies have utilized this method to assess cell migration and proliferation [67,68,69,70,71]. Monolayers of HUVECs were pretreated with D-ALG or ALG-1001 followed by scratching the monolayer with a pipette tip. Images were taken at the time of the wound and at 24 h after injury. The proliferative activity of endothelial cells can be measured by the area that was healed postinjury to the monolayer. Untreated cells managed to heal up to 80% of the initial damage after 24 h, while cells treated with D-ALG and ALG-1001 healed up to 50% of the initial wound area. Further, cells treated with a high dose of D-ALG at 1 mM only recovered 20% of the initial damage, suggesting a reduced ability of HUVECs to regenerate (Appendix A). Because neovascularization is driven in part by the proliferation and migration of endothelial cells in the disease area, reductions in these activities as observed in cells treated with D-ALG can modulate endothelial cell contributions to the neovascularization process. Further, comparing the efficacy between D-ALG and free ALG-1001 suggests that treatment with D-ALG is more efficacious than the free peptide at reducing neovascularization-related cellular processes.

### 3.5. Western Blot on Endothelial Cell Activation

To elucidate the mechanism through which D-ALG and free ALG-1001 influence angiogenic function, we pretreated the cells with D-ALG and free ALG-1001 for 24 h. The cells were then stimulated with a high dose of exogenous VEGF for 5 min after which the samples were collected for Western blotting. We probed for the expression of total FAK, phospho-FAK (Y397), ERK1/2, phospho-ERK1/2, and cyclophilin B (CycB) with CycB acting as the internal control.

Compared to untreated, VEGF stimulated controls, the trend suggested that cells treated with D-ALG and ALG-1001 expressed lower levels of phospho-ERK1/2 and phospho-FAK at doses of 100 µM on a peptide basis. Both ERK1/2 and FAK pathways are important players in angiogenesis and their activation promotes endothelial cell proliferation and migration (Figure 4). The trend in the reduction in phosphorylated proteins suggests an attenuation in the activation of these networks in response to VEGF.

In addition to examining pathway activation, we also studied the effects of D-ALG and ALG-1001 on VEGF-A expression by endothelial cells (Appendix A). Cells treated with VEGF-A significantly increased the production of VEGF-A compared to unstimulated controls. Cells coincubated with VEGF-A and either free ALG-1001 or D-ALG did not have statistically significant increases in VEGF production compared to the control, suggesting a minor attenuation in endothelial activation in response to VEGF.

### 3.6. Attenuation of Inflammatory Response in Murine Microglia

Other important players in angiogenesis are macrophages and microglia which produce proinflammatory and proangiogenic cytokines during the angiogenic process. To elucidate the effect of D-ALG and ALG-1001 on macrophage activation, we pretreated RAW264.7 macrophage cells with a high and low dose of D-ALG and free ALG-1001 24 h prior to LPS stimulation. The treatment media was first aspirated to simulate the transient nature of in vivo delivery, and cells were activated with LPS for 3 h.

Cells activated with LPS displayed a robust increase in the expression of proinflammatory cytokines IL1β and TNFα as detected through qPCR when compared to unstimulated controls (Figure 5). With D-ALG treatment, the expression of IL1β was reduced by ~90% and TNFα by 80% in both the low (100 µM) and high dose (1 mM). In comparison, ALG-1001 reduced the expression of TNFα only by a relatively moderate 20%, only at the highest dose as well. We observed no impact on IL1β production with free ALG-1001 treatment alone. This major difference in the efficacy between free ALG-1001 and the dendrimer conjugate may be due to two potential differences: (1) the dendrimer platform is demonstrated to be much more efficient at delivering therapeutics to activated macrophages compared to free drugs; (2) attaching multiple ligands on the dendrimer surface may allow for multivalency effects, increasing the interaction of the peptide and surface integrin.

### 3.7. Biodistribution of Systemically Administered ALG-1001 and D-ALG in Laser CNV Model

Fluorescently labeled Cy3-ALG-1001 peptide and dual-labeled Cy5-dendrimer-ALG-Cy3 were systemically injected on day 0 (same day as CNV induction in the laser CNV model) and choroidal tissues were collected at set time points. Confocal microscopy was used to monitor the presence of ALG-1001 peptide (Cy3), dendrimer carrier (Cy5), and CNV formation (isolectin).

Within the first 24 h of systemic administration, free ALG-1001 peptide reached the CNV area and remained there for up to 2 days as seen from detected Cy3 signals (Figure 6). In comparison, D-ALG was able to not only reach the CNV area within 24 h after systemic injection but also remain in the target area up to at least 4 days postadministration from the colocalized presence of both Cy5 and Cy3 signals. The prolonged residence time may suggest that dendrimer conjugation allowed the target area itself to serve as a drug depot, prolonging the efficacy of the peptide therapeutic. This prolonged residence time may be due to a better resistance to enzymatic degradation, and/or due to cellular uptake into the resident macrophages in the target area that is an intrinsic property of dendrimers.

### 3.8. Systemically Administered D-ALG Attenuates FAK and ERK Activation

To evaluate the activation of FAK and ERK pathways, eyes were enucleated at set time points, and the choroidal tissue dissected out. Tissues were submerged in a mixture of T-per, proteinase inhibitor, and PhosStop with stainless-steel homogenization beads and homogenized to produce protein extracts. Total FAK, total ERK, p-FAK (Y397), and p-44/42 ERK ELISA kits were used to quantify the amount of total and phosphorylated proteins in FAK and ERK pathways. In animals treated with D-ALG, a trend in the reduction in total FAK and p-FAK proteins was observed over both 7 and 14 days compared to untreated controls, suggesting a prolonged attenuation of the FAK pathway (Figure 7A). The treatment of ALG-1001 also resulted in a trend in the reduction in p-FAK across both time points. However, the level of total FAK protein in ALG-1001-treated animals was elevated at day 14. Both D-ALG- and ALG-1001-treated animals resulted in a similar trend in the decrease in p44/42 ERK production, while total ERK remained relatively constant at day 7 across treatment groups. On day 14, there was a slight trend in the reduction in total ERK in D-ALG- and ALG-1001-treated animals compared to untreated animals.

To compare the expression of proinflammatory and proangiogenic cytokines, RNA was extracted, and qPCR was performed to determine the relative expression of VEGF-A, TNFα, and IL1β (Figure 7B). In D-ALG treated animals, we observed a trend that VEGF-A production was reduced over both 7 and 14 days, whereas free ALG-1001 treated animals expressed lower VEGF-A production at 14 days only. Comparing trends in inflammatory cytokines, D-ALG produced lower TNFα level at 7 days only, while ALG-1001-treated animals had lower TNFα levels across both time points. The trend of IL1β expression was only reduced at day 14 for ALG-1001- and D-ALG-treated animals.

### 3.9. In Vivo Attenuation of CNV Formation

A Micron III SLO scope and a laser attachment were used to induce CNV formation in mice. The laser CNV model was chosen due to its consistency in creating CNV and the progression of the CNV process [72,73]. We first evaluated whether dendrimer conjugation impeded the peptide from attenuating CNV by first injecting free ALG-1001 and D-ALG intravitreally after CNV induction. The eyes were then collected on day 7 and the CNV area quantified. Both D-ALG and ALG-1001 significantly inhibited the formation of CNV when administered intravitreally (Appendix A).

We then evaluated whether the improved protection and targeting of D-ALG allowed for less invasive administration routes. A laser was used to induce CNV at day 0, and the first dose (150 µg peptide basis) of ALG-1001 and D-ALG was administered intraperitoneally. The animals were dosed once every 4 days at 150 µg of peptide basis (7.5 mg/kg). Choroidal flatmounts of eyes at day 7 and 14 were imaged, and the CNV area was calculated using ImageJ.

In untreated control animals, the CNV area reached a size of 13,000 µm^2^ by day 7 and 14,000 µm^2^ by day 14 after CNV induction (Figure 8). The systemic injection of ALG-1001 produced a robust attenuation of the CNV formation (~60% reduction) on day 7, but the CNV area recovered at day 14. In comparison, the systemic administration of D-ALG suppressed the formation of CNV by ~50% both on day 7 and day 14, with day 14 reaching statistical significance. This improvement in CNV reduction and its sustained effect up to 14 days, following a single intraperitonial administration of D-ALG-1001, is a significant advance and may be attributed to the ability of D-ALG to protect the peptide payload and lengthen the residence time at the target CNV area [74].

## 4. Discussion and Conclusions

Peptide drugs face many challenges relating to plasma stability and intracellular delivery to target cells, which has hampered their clinical translation. We used hydroxyl PAMAM dendrimers, which have shown potential in phase 2 trials, to deliver peptides to areas of neovascularization from systemic administration. Dendrimer conjugation of ALG-1001 peptide was accomplished through a highly efficient copper-assisted click reaction under mild conditions and characterized through HPLC and NMR. Through the integration of NMR peaks, we calculated that 6–7 peptides were attached per dendrimer carrier (~6 wt% payload). We observed an increased resistance to enzymatic degradation with dendrimer conjugation in vitro after incubation with a broad-acting proteinase, where the conjugated peptide was stable up to at least 90 min in plasma.

At high, equivalent doses, D-ALG conjugates inhibited vessel formation an order of magnitude better than the free peptide. In addition, D-ALG1001 attenuated not only the activation of the FAK pathway in endothelial cells but also the expression of proinflammatory cytokines in LPS-stimulated murine macrophages. We hypothesize that attaching multiple peptide moieties on a single dendrimer allowed the integrin clusters to be engaged more effectively, enhancing the antiangiogenic and anti-inflammatory activity of ALG-1001 through this multivalency effect.

When systemically administered in vivo, differences in the distribution and bioavailability of D-ALG and free ALG-1001 peptide could be better elucidated. Fluorescently labeled D-ALG was detectable in the CNV area up to 4 days after intraperitoneal injection; on the other hand, free ALG-1001 signal was undetectable after day 2. The increased residence time at the target area allowed for less frequent injections for D-ALG. An intraperitoneal injection of 150 µg of D-ALG (on a peptide basis) every four days resulted in a 50% reduction in the CNV area, whereas free ALG-1001 peptide reduced CNV formation by 40% on day 7 and lost potency at later time points (20% CNV reduction on day 14 for free ALG-1001 vs. 50% CNV reduction on day 14 for D-ALG). In comparison, the continuous systemic delivery of JSM6427, a small molecule α5β1 antagonist, using an implanted pump resulted in a 40% CNV area reduction [75]. Similarly, in separate studies by Das et al. and Toriyama et al., exploring Å6 peptide and CGRP peptides, respectively, daily injections were necessary to reduce CNV area by 30% [76,77].

The work presented here demonstrates that dendrimer conjugation can not only deliver biologics intact to the target area after systemic administration but also increase its residence time and efficacy. As a result, a less stringent dosing schedule is needed to effectively control CNV formation. The work presented here suggests that dendrimer conjugation is a valid strategy to significantly improve the stability of biologics and as a vehicle for the systemic, targeted delivery of these powerful therapeutics.

## Figures and Tables

**Figure 1 pharmaceutics-15-02428-f001:**
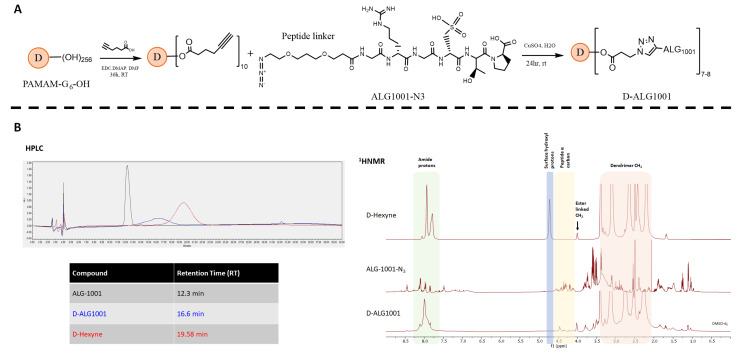
Synthesis and characterization of D-ALG1001. (**A**) Synthesis scheme of D-ALG1001: surface of generation-6 hydroxyl-terminated dendrimer was functionalized with alkyne-terminated linkers. ALG-1001 peptide was then attached using a copper-catalyzed click reaction to yield D-ALG conjugates. (**B**) The D-ALG conjugate was characterized by HPLC (**left**) with a retention time of 16.6 min compared to precursor materials ALG-1001 (12.3 min) and D-hexyne (19.58 min) with the chromatogram trace color corresponding to the text color. ^1^H NMR characterization of the precursor and final conjugate (**right**) shows the appearance of characteristic signals.

**Figure 2 pharmaceutics-15-02428-f002:**
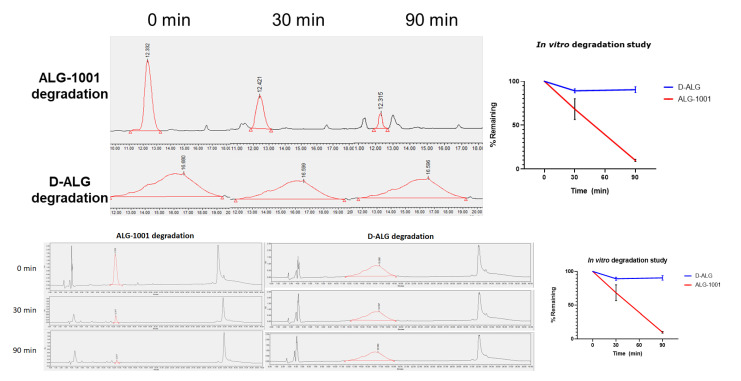
In vitro enzymatic degradation in the presence of proteinase K. HPLC chromatographs of D-ALG and ALG-1001 after coincubation with proteinase K show a decrease in the AUC of the peak associated with the free ALG-1001 peptide (**top**). The trace of D-ALG shows a negligible decrease in AUC (**middle**). The peaks corresponding to ALG-1001 and D-ALG are outlined in red. The curve plotting the amount degraded is shown by normalizing peaks of analyte taken at set time points to the starting peak obtained at 0 min (**bottom**). About 90% of ALG-1001 was degraded by 90 min whereas only 10% of D-ALG was degraded.

**Figure 3 pharmaceutics-15-02428-f003:**
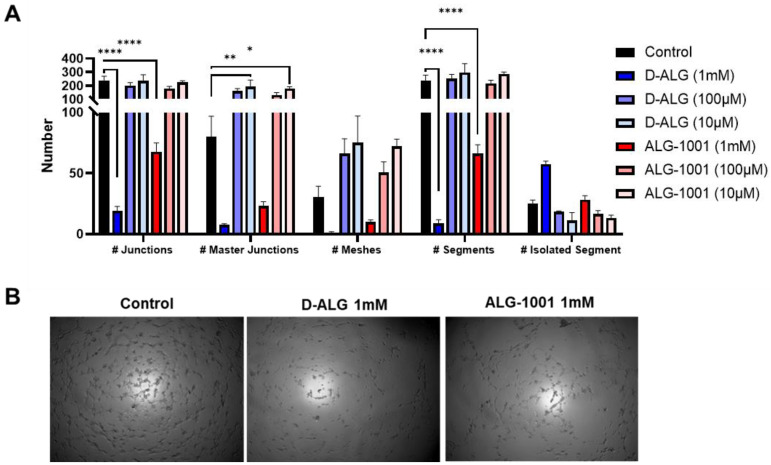
In vitro inhibition of vessel formation. (**A**) Image analysis of HUVECs seeded onto a Matrigel matrix using the Angiogenesis Analyzer plug-in for ImageJ. The relevant metrics extracted for the integrity and expanse of vessel formation were the number (#) of times vessels intersected one another (junctions), the number (#) of spaces enclosed by vessels (meshes), the number (#) of connected vessels (segments), and the number (#) of isolated vessels (isolated segments). Significant disruptions in vessel formation resulted in reduced junctions, segments, and meshes while the number of isolated segments increased. Cells treated with 1 mM ALG-1001 and D-ALG significantly reduced vessel formation compared to untreated cells. * *p* < 0.05, ** *p* < 0.01, and **** *p* < 0.0001 (**B**) Representative images of vessel formation from untreated cells (control), and cells treated with D-ALG and ALG-1001 at a 1 mM concentration.

**Figure 4 pharmaceutics-15-02428-f004:**
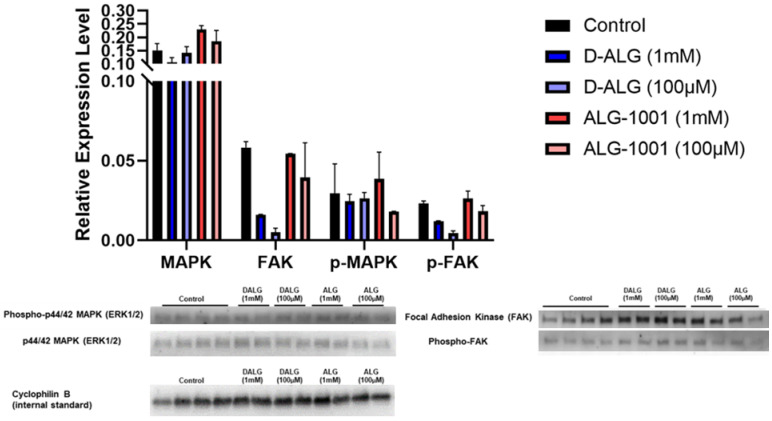
Activation of ERK and FAK pathway in response to VEGF stimulation. Relative protein expression was calculated by analyzing protein bands associated with ERK (42 and 44 kDa) and FAK (110 kDa) and normalizing the content to an internal control (cyclophilin B) (**top**). Representative images of blots are shown (**bottom**).

**Figure 5 pharmaceutics-15-02428-f005:**
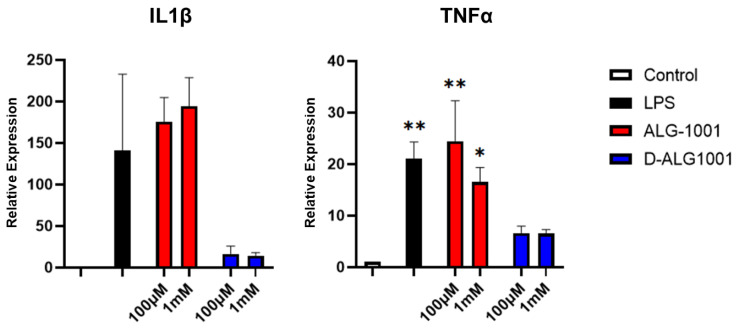
Expression of proinflammatory cytokine of RAW264.7 cells in response to LPS stimulation. Compared to untreated controls, RAW cells stimulated with LPS produced robust expressions of IL1β and TNFα. Pretreatment with ALG-1001 produced a slight effect at attenuating TNFα expression at 1 mM concentration; on the other hand, treating with D-ALG brought the level of TNFα to a level not statistically different than untreated controls. *p*-values denoted here compare the level of IL1β and TNFα expression to untreated controls. * *p* < 0.05, ** *p* < 0.01.

**Figure 6 pharmaceutics-15-02428-f006:**
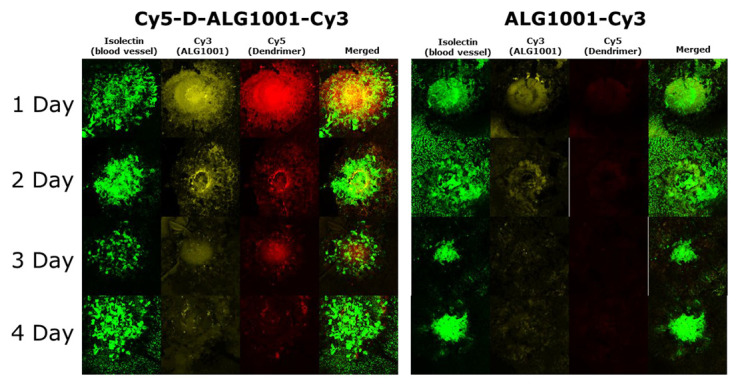
Biodistribution of intraperitoneally injected Cy5-D-ALG-Cy3 and ALG-1001-Cy3 at areas of choroidal neovascularization. CNVs were stained with isolectin (green), ALG-1001 was labeled with Cy3 (yellow), and the dendrimer was labeled with Cy5 (red). Cy5-D-ALG-Cy3 conjugates stayed in the target area over a period of 4 days with the signal staying around the CNV area. A majority of ALG-1001-Cy3 stayed in the target area over 2 days, and most were cleared by day 3.

**Figure 7 pharmaceutics-15-02428-f007:**
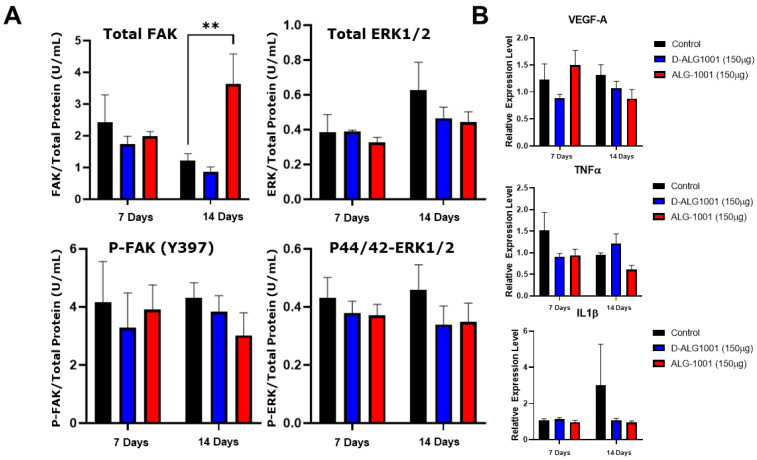
Quantification of ERK and FAK pathway activation and expression of proinflammatory/angiogenic cytokines. (**A**) Quantification of FAK, phosphor-FAK (Y397), p44/42 ERK, phospho-p44/42 ERK proteins using ELISA kits. Trends indicate a reduced phosphorylation of p44/42 ERK over both 7 and 14 days compared to controls in both treatment groups. In D-ALG-treated animals, there is also a trend in reduced phosphor-FAK protein expression across both time points. ** *p* < 0.01 (**B**) Quantification of VEGF-A, TNFα, and IL1β through qPCR. The trend shows a reduction in VEGF-A and TNFα at day 7 for animals treated with D-ALG and a reduction in IL1β production on day 14. ALG-1001-treated animals had a trend of lower TNFα at both time points and lower VEGF-A and IL1β on day 14.

**Figure 8 pharmaceutics-15-02428-f008:**
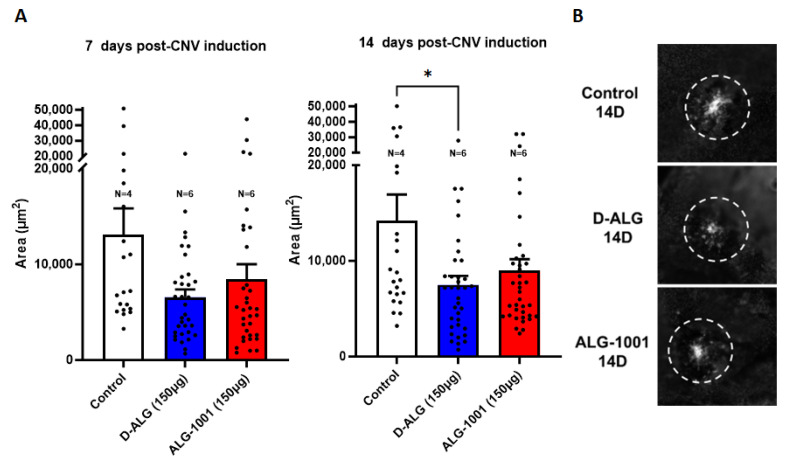
In vivo inhibition of CNV formation in a laser CNV mouse model, following intraperitonial administration of free ALG-1001 and D-ALG-1001. (**A**) Area of CNV as quantified using ImageJ. The trend suggests D-ALG- and ALG-1001-treated animals had smaller CNV areas compared to untreated animals at day 7. The degree of CNV reduction is retained in D-ALG-treated animals on day 14 and the difference reaches statistical significance. (**B**) Representative images of CNV on day 14. * *p* < 0.05.

## Data Availability

The data presented in this study are available on request from the corresponding author.

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
