# Peer review of "Systemic Dendrimer-Peptide Therapies for Wet Age-Related Macular Degeneration"

_pharmaceutics, 2023, doi:10.3390/pharmaceutics15102428_

Round 1
Reviewer 1 Report
In this manuscript, the authors presented a novel dendrimer-integrin binding peptide (D-ALG) which can treat AMD through systemic administration. Compared with free ALG-1001, D-ALG has better anti-degradation ability and can maintain higher local drug concentration, which greatly improves drug bioavailability. I recommend this manuscript to be accepted after addressing the following issues.
1. In wound healing assays, high doses of D-ALG work better, but low doses of ALG-1001 work better. Why are the results different for ALG-1001 and D-ALG?
2. In Western blot on endothelial cell activation, it is mentioned that "Compared to untreated, VEGF stimulated controls, the trend suggests that cells treated with D-ALG and ALG-1001 expressed lower levels of total FAK and phosphor-FAK." However, as shown in Figure 4, compared with the control group, ALG-1001 at high doses of 1mM did not significantly reduce FAK levels, and p-FAK levels were higher.
3. Does D-ALG have an effect on the activity of normal cells in the eye compared to ALG-1001, and how is the cytotoxicity of D-ALG?
4. In Figures S1-S6, it is recommended to label the molecular structure next to the peaks instead of lines. This will make the results more intuitive and convenient.
Reviewer 2 Report
The authors report on dendrimer conjugation to improve delivery of integrin inhibitors for the treatment of neovascularization in retina. They conclude that this process decreases the enzymatic degradation and increases the stability of the compound. There seems to be a dose-escalation effect on vascular disruption in vitro and a healing and anti-inflammatory effects. The experimental methodology is solid and well-described. As this approach for retinovascular disease has been already tried in Phase 1 clinical studies it is a promising approach and any additional lab studies are contributory.
Comments:
Were there any cell/tissue toxicity studies done or previously reported?
Perhaps more anti-inflammatory cytokines could have been used to assess anti-inflammatory effect. Inhibition of IL1-beta and TNF-alpha may not give a full picture of anti-inflammatory effects.
Please standardize font in the Statistics section.
Reviewer 3 Report
A novel dendrimer modified integrin binding peptide was synthesized and showed good in vitro and in vivo anti-angiogenic activities. There is no revision comment in the present form.
Author Response
No revisions. Thanks!
Round 2
Reviewer 1 Report
The authors have addressed all my concerns.